bioengineering

ventilator splitting, ventilator sharing, COVID-19, coronavirus, acute respiratory distress syndrome, ARDS

**Author for correspondence:**
Steven E. Williams
e-mail: steven.e.williams@kcl.ac.uk

# A simulated single ventilator/dual patient ventilation strategy for acute respiratory distress syndrome during the COVID-19 pandemic

José A. Solís-Lemus[1], Edward Costar[2], Denis Doorly[3], Eric C. Kerrigan[3,4], Caroline H. Kennedy[5], Frances Tait[6], Steven Niederer[1], Peter E. Vincent[3] and Steven E. Williams[1,7]

[1]School of Biomedical Engineering and Imaging Sciences, King's College London, London WC2R 2LS, UK
[2]Imperial College Healthcare NHS Trust, UK
[3]Department of Aeronautics, and [4]Department of Electrical and Electronic Engineering, Imperial College London, UK
[5]Evelina Children's Hospital, Guy's and St Thomas' NHS Foundation Trust, UK
[6]University Hospitals of Leicester NHS Trust, UK
[7]Department of Cardiology, Guy's and St Thomas' NHS Foundation Trust, UK

JAS-L, 0000-0002-2596-7811; SN, 0000-0002-4612-6982

The potential for acute shortages of ventilators at the peak of the COVID-19 pandemic has raised the possibility of needing to support two patients from a single ventilator. To provide a system for understanding and prototyping designs, we have developed a mathematical model of two patients supported by a mechanical ventilator. We propose a standard set-up where we simulate the introduction of T-splitters to supply air to two patients and a modified set-up where we introduce a variable resistance in each inhalation pathway and one-way valves in each exhalation pathway. Using the standard set-up, we demonstrate that ventilating two patients with mismatched lung compliances from a single ventilator will lead to clinically significant reductions in tidal volume in the patient with the lowest respiratory compliance. Using the modified set-up, we demonstrate that it could be possible to achieve the same tidal volumes in two patients with mismatched lung compliances, and we show that the tidal volume of one patient can be manipulated independently of the other. The results indicate

that, with appropriate modifications, two patients could be supported from a single ventilator with independent control of tidal volumes.

## 1. Background

COVID-19 is a viral illness caused by the recently discovered coronavirus, severe acute respiratory syndrome coronavirus 2 (SARS-CoV-2). The disease originated in Wuhan, Hubei Province, China and was first documented through a series of unexplained pneumonia cases in December 2019 [1]. The disease subsequently spread rapidly worldwide and by the end of February 2020 several countries, including in Europe, were experiencing sustained transmission. According to the WHO as of 2 April 2020, there were 900 306 confirmed cases worldwide in 206 countries, areas or territories and 45 693 confirmed deaths [2]. The virus continued to spread globally with a reproductive number estimated at 2.5. COVID-19 causes a multitude of symptoms with the main symptoms seen being fever and dry cough. Although some cases are asymptomatic, severe disease can cause death at an estimated rate of around 3–4% [2].

Although myocardial damage and circulatory failure contribute to COVID-19 deaths, the main cause of death is respiratory failure [3] and the majority of serious cases require intensive care unit (ICU) admission and mechanical ventilation. It is estimated that the ICU capacity of all EU/EEA countries (including the UK) would be exceeded at a prevalence of 100 hospitalized COVID-19 patients per 100 000 of the population (based on the Hubei Province scenario at the peak of the epidemic) [4]. According to the most recent European Centre for Disease Prevention and Control report, the majority of EU/EEA countries were predicted to reach this scenario by the end of March 2020 [4]. Major difficulties in predicting the course of the outbreak given the exponential spread during the early phase, mitigated by behavioural changes and government methods, mean there is large uncertainty in the models, but widespread agreement that ventilator availability is likely to be a critical factor in patient care. As such, there is a worldwide concern that there will be a shortfall of mechanical ventilators during peaks of this global pandemic. As one example, estimates of the number of ventilators in the USA range from 60 000 to 160 000, while up to 1 million ventilators may be required at the height of the USA pandemic [5]. Regardless of the strategy used for estimating this latter number, the national strategic reserve is not thought to be sufficient to fill the projected gap [6].

Given the predictions for a huge global shortfall in ventilators, strategies have been proposed for ventilator sharing. In 2006, Neyman & Babcock [7] found that a single ventilator could be modified quickly in an emergency department setting to ventilate four simulated adult patients for a short period of time. Subsequently Paladino et al. [8] successfully ventilated four adult-human-sized sheep on a single ventilator for at least 12 h. During the early stages of the COVID-19 pandemic, this approach has received significant media attention [9], with at least two clinical protocols in development [10], and it is reported that national decision makers are either considering [11] or have recommended this approach [12].

Importantly, such ventilation strategies assume equal lung physiology in all patients and are likely only to be successful in situations where multiple patients with similar lung physiology require ventilation. During COVID-19, the clinical picture can range from a mild illness to pneumonia, severe pneumonia, acute respiratory distress syndrome (ARDS), sepsis and septic shock [13]. ARDS is an acute diffuse, inflammatory lung injury, leading to increased pulmonary vascular permeability, increased lung weight and loss of aerated lung tissue [14]. It is a heterogeneous disease that lowers lung compliance as a function of disease severity and it is highly unlikely that any two COVID-19 patients will have either the same lung physiology or the same ventilator requirements.

As such, the practice of ventilator sharing has been widely debated. In the USA, the Society of Critical Care Medicine (SCCM), American Association for Respiratory Care (AARC), American Society of Anesthesiologists (ASA), Anesthesia Patient Safety Foundation (APSF), American Association of Critical-Care Nurses (AACN) and American College of Chest Physicians (CHEST) released a joint statement in March 2020 advising against ventilating multiple patients per ventilator (while any other clinically proven, safe and reliable therapy remains available) [15]. This statement cited multiple problems with ventilator sharing including (i) the inability to deliver different pressures or achieve the required tidal volumes in individual patients, (ii) the inability to manage positive end-expiratory pressure (PEEP), which is of critical importance to these patients, (iii) risks of cross-infection, (iv) difficulties monitoring both patients simultaneously, (v) difficulties arising from one patient deteriorating suddenly or having a cardiac arrest, and (vi) ethical issues [15]. To be absolutely clear, the ventilation of two patients using a

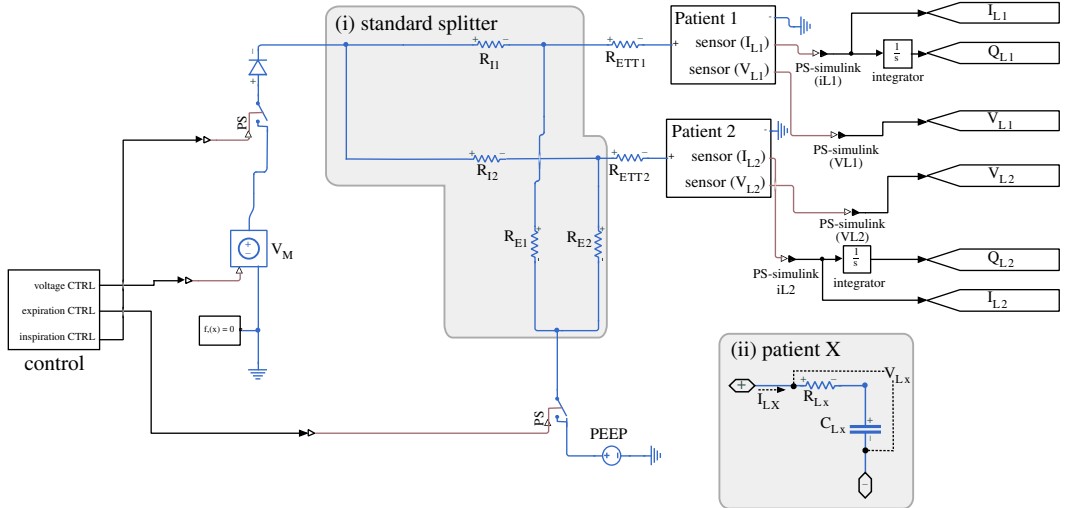

**Figure 1.** Circuit diagram for the standard splitter. The control module contains a pulse-wave generator with a period appropriate to the respiration rate. It delivers three square signals to control the inspiration, expiration and voltage to the circuit. The $R_{ETT}$ resistors correspond to the endotracheal tubing. $R_{Ex}$ and $R_{Ix}$ correspond to the inflow and outflow tubing resistance. The patient module consists of a resistor ($R_{Lx}$) to model upper airway resistance, and a capacitor $C_{Lx}$, to model the compliance of the lungs and chest wall. Grey boxes outline the splitter (i) and patient submodule (ii). The text 'sensor' on the patient blocks indicate that the respective signals are connected to appropriate Simscape sensor blocks, namely current and voltage sensors, to allow for these signals to be logged during simulation.

single machine is *strongly advised against*. However, given the current unprecedented situation, it appears likely that there will not be enough ventilators worldwide, therefore leading to indirect deaths due to lack of available resources.

In this simulation study, we therefore develop, validate and make freely available a model of a mechanical ventilator supporting two patients using a simple T-junction splitter to supply air to both patients, based on the design proposed by Neyman & Babcock [7]. We use the model to simulate ventilating two patients with different degrees of lung disease from a single ventilator and estimate the delivered volumes and pressures in each patient. This provides a quantitative estimate of the potential risks involved in this strategy. We then propose the addition of variable resistances and one-way valves to the ventilator tubing system and show that in our model these can be used to regulate tidal volumes in each patient independently in the presence of mismatched respiratory physiology.

## 2. Methods

We designed an electrical circuit as an analogue to the ventilator, the system connecting the ventilator to the two patients and the lungs of each patient [16]. In mapping from the air flow to an electrical circuit, we equate volume to charge, flow rate to current, pressure to voltage, resistance to resistance and compliance to capacitance. In the ventilator model, this means that tubes appear as resistors, the lung and chest wall as a capacitor, the pressure source as a voltage source, a one-way valve as a diode and an open/closed valve as a switch.

Specifically, we designed two models. The first was a simple *Standard Splitter* configuration that involved connecting two T-junctions to the ventilator, as proposed previously [7]. This configuration is shown in figure 1. The second was a *Modified Splitter* configuration, which had two elements added to each branch of the splitter: specifically, on the inspiration arms we added variable resistors ($R_{V1}$ and $R_{V2}$) to control pressure/flow/volume to each patient, and on the expiration arms, we added diodes to prevent backflow through each of the channels and stop the expiration arms acting as a short circuit between inspiration arms during inspiration. This configuration is shown in figure 2. The model outputs the pressure set by the ventilator ($V_M$) and the pressure ($V_{L1}$ and $V_{L2}$), flow ($I_{L1}$ and $I_{L2}$) and volume ($Q_{L1}$ and $Q_{L2}$) delivered to each patient. The ventilator pressure set by $V_M$ is defined as a square wave. The maximum value is set to the peak inspiration pressure (PIP). The minimum value is set to PEEP. The cycle rate is defined by the respiratory rate (RR) measured in breaths per minute. The ratio of time spent at the maximum and minimum values is defined by the inspiration to expiration ratio (I : E ratio).

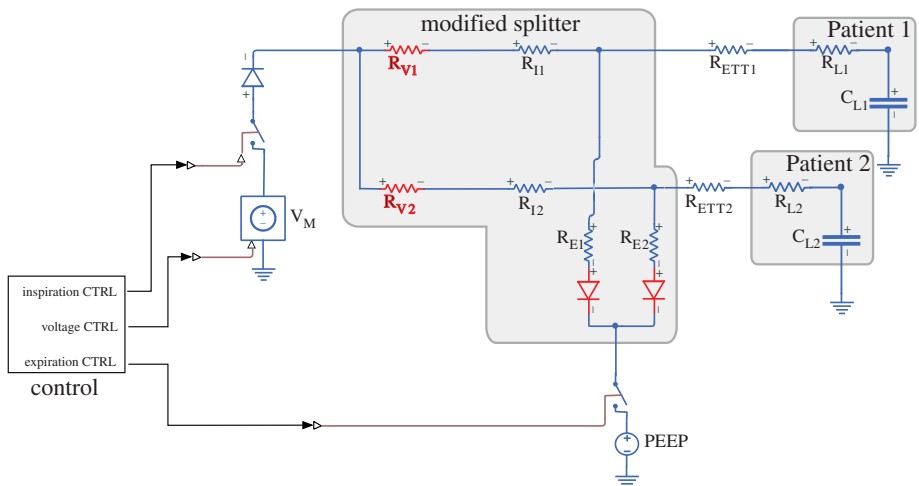

**Figure 2.** Circuit diagram for the modified splitter. Compared to figure 1, variable resistors $R_{V1}$ and $R_{V2}$ have been added to control the pressure/flow/volume to each patient, and diodes have been added to stop the expiration arms acting as a short circuit between inspiration arms during inspiration. Changes with respect to the standard circuit are highlighted in red.

**Table 1.** Parameters of the model in units from clinical practice converted to S.I. units.

| parameter | clinical units | S.I. Units | references |
|---|---|---|---|
| *ventilator parameters* | | | |
| PEEP | 5 cmH₂0 | 490 Pa | [17,18] |
| PIP (reference value) | 15 cmH₂0 | 1471 Pa | [17,18] |
| RR | 15 breaths/min | | [17,18] |
| I : E ratio | 1 : 2 | | [17,18] |
| *splitter parameters* | | | |
| $R_I$ | 0.06 cmH₂0/L/s | 5670 Pa s m$^{-3a}$ | derived |
| $R_V$ | variable | variable | N/A |
| $R_E$ | 0.06 cmH₂0/L/s | 5670 Pa s m$^{-3a}$ | derived |
| *patient parameters* | | | |
| $R_L$ | 2 cmH₂0/L/s | 196133 Pa s m$^{-3}$ | [19] |
| $C_L$ (reference value) | 0.054 L/cmH₂0 | $5.5 \times 10^{-7}$ m$^3$ Pa$^{-1}$ | [17] |
| $R_{ETT}$ | 8.0 cmH₂0/L/s | 784000 Pa s m$^{-3}$ | [19] |

[a]Resistance (R) derived from Poiseuille law for straight laminar pipe, $R = (128\,\mu L)/(\pi D^4)$, assuming diameter $D = 22$ mm, viscosity of air $\mu = 18.13 \times 10^{-6}$ Pa s, length $L = 1.8$ m. PEEP, positive end-expiratory pressure; PIP, positive inspiratory pressure; RR, respiratory rate; I : E ratio, inspiratory/expiratory ratio. Reference values represent healthy patients and are used for Lung Model A, defined below.

## 2.1. Definition of parameters

We defined the parameters for both the standard and modified splitter configuration in S.I. units, based on the literature. Specifically, there are 16 parameters subcategorized into ventilator parameters, splitter parameters and patient parameters. The units used in clinical practice were changed to S.I. units in the model (table 1).

To simulate ARDS in the model, the values of $C_L$ were varied assuming 0.054 L/cmH₂O corresponds to a compliance of 100%. We defined four models to represent patient types with increasing severity of ARDS with decreasing compliances as follows: Lung Model A (healthy, compliance 0.054 L/cmH₂O), Lung Model B (mild ARDS, 20% compliance reduction, 0.0432 L/cmH₂O), Lung Model C (moderate ARDS, 30% compliance reduction, 0.0378 L/cmH₂O) and Lung Model D (severe ARDS, 40%

compliance reduction, 0.0324 L/cmH$_2$O). The reduction in compliance for Lung Models B, C and D were chosen to represent a range of compliances around the reported median compliance for patients with ARDS [17].

## 2.2. Implementation

We implemented the simulations using MathWork's Simscape (Simulink v4.8) Foundational Blocks and the tests were run via Matlab R2020a scripts and functions. All code is freely available on GitHub (https://github.com/splitvent/splitvent) under an MIT licence. Simulations were initialized with all state variables set to 0. Simulations visually reached quasi-steady equilibrium after the first pulse and all reported results are taken from the second pulse onwards. The default Simulink and Simscape settings were used to generate the data, namely automatic selection of solver and step sizes with a relative error tolerance of $10^{-3}$. Simulink/Simscape was able to reduce the model to an ordinary differential equation (ODE) and we chose to solve the ODE with the Dormand-Prince explicit, variable-step RungeKutta (4,5) formula (the Matlab ode45 solver), with a maximum step size of 0.3. The plots were generated using the resulting variable-step size data, but joined up with lines.

## 2.3. Experiments

We designed four experiments to answer the research questions for this work:

(i) *Model validation.* Using the standard splitter, we connect patients that have lungs with identical parameters, i.e. Lung Models A–A, B–B, C–C and D–D, and ventilate using pressure control ventilation.

(ii) *Evaluating the standard splitter when ventilating two patients with mismatched lung compliance.* Using the standard set-up, we connected patients with different levels of respiratory compliance, i.e. Lung Models A–B, A–C and A–D, to quantify the effect of delivering the same ventilator settings to two patient with mismatched lung compliance.

(iii) *Achieving equivalent tidal volumes in patients with mismatched lung compliance.* Using the modified splitter, we connected patients with different levels of respiratory compliance, i.e. Lung Models A–B, A–C and A–D, to demonstrate whether, by adjusting the values of $R_{V1}$, $R_{V2}$ and PIP, we are able to deliver equal tidal volumes to two patients with mismatched respiratory compliance.

(iv) *Independently adjusting tidal volumes in both patients.* In two patients, both with Lung Model C (moderate ARDS), we aimed to achieve a 30% increase in tidal volume or a 30% decrease in tidal volume in one patient while maintaining the tidal volume in the other patient by adjusting $R_{V1}$, $R_{V2}$ and PIP.

# 3. Results

## 3.1. Model validation

To validate the model simulations, pairs of patients with identical lung models were ventilated using the standard splitter. Simulations of pressure control ventilation for pairs of patients with healthy lungs (i.e. Lung Model A + Lung Model A (A–A)) and pairs of patients with ARDS lungs (i.e. Lung Model B + Lung Model B (B–B), Lung Model C + Lung Model C (C–C) and Lung Model D + Lung Model D (D–D)) were performed (figure 3a). The initial ventilator settings were PIP 15 cmH$_2$O, PEEP 5 cmH$_2$O, RR 15 breaths/min, I : E ratio 1 : 2. For case A–A (healthy lungs), a tidal volume of 490 ml was achieved, which equates to 6.2 ml kg$^{-1}$ ideal body weight for a 185 cm male [18]. The respiratory pressure curves (an estimate of alveolar pressure measured at the patient end of the simulated endotracheal tube, i.e. at the patient side of $R_{ETT}$) are shown in (figure 3b). As expected, we demonstrate a progressive reduction in tidal volume as respiratory compliance decreases (figure 3b, black arrow). Achieved PEEP was 5 cmH$_2$O in each case. We then showed that tidal volume could be returned to normal for each Lung Model by increasing PIP. For the ARDS cases (B–B, C–C and D–D) PIP was increased to achieve a tidal volume within 10 ml of 490 ml (the tidal volume seen in Lung Model A). Increasing PIP to 15.16 cmH$_2$O, 17 cmH$_2$O and 18.5 cmH$_2$O for cases B–B, C–C and D–D, respectively, normalized tidal volume for both patients (table 2)

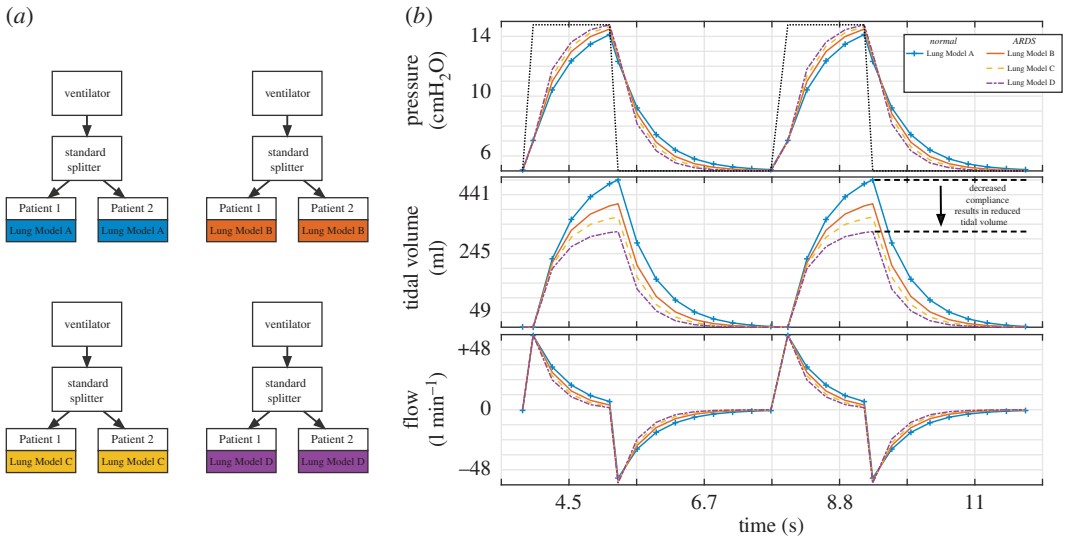

**Figure 3.** Comparison of pressure–time, volume–time and flow–time curves for Lung Models A–D. (*a*) In this experiment, two patients are connected via the standard splitter to the ventilator. For each simulation, each pair of patients was parametrized with the same Lung Model. (*b*) Ventilation was performed in pressure control mode with PIP 15 cmH₂O, PEEP 5 cmH₂O, RR 15 breaths/min, I : E ratio 1 : 2 (dotted line, top panel). Pressure–time, tidal volume–time and flow–time graphs are shown for Patient 1. Note the progressive reduction in tidal volume from Lung Model A through D (black arrow, middle panel). Identical pressure–time, tidal volume–time and flow–time graphs were seen for Patient 2 (not shown).

**Table 2.** PIP and tidal volumes for simulation of paired equivalent patients with increasing lung disease represented by decreasing lung compliance. PIP was adjusted to achieve a tidal volume of approximately 490 ml.

| Patient 1–Patient 2 | PIP (cmH₂O) | Patient 1 tidal volume (ml) | Patient 2 tidal volume (ml) |
|---|---|---|---|
| A–A | 15 | 490.2 | 490.2 |
| B–B | 17 | 493.2 | 493.2 |
| C–C | 18.5 | 494.5 | 494.5 |
| D–D | 20.5 | 493.6 | 493.6 |

## 3.2. Quantifying the pressure and tidal volumes when ventilating two patients with different degrees of lung disease

To provide a quantitative estimate of the effect of the standard unregulated splitter on the tidal volume delivered to each patient, we simulated the extreme case of pairing a patient with healthy lungs with patients with increasing severity of ARDS manifested by decreasing lung compliance. Simulations of pressure control ventilation for pairs of patients with the following lung models were performed: Lung Model A + Lung Model A (A–A), Lung Model A + Lung Model B (A–B), Lung Model A + Lung Model C (A–C) and Lung Model A + Lung Model D (A–D) (figure 4*a*). In all simulations, the PIP was set to achieve that required to successfully ventilate the normal lung model (15 cmH₂O) and the other ventilator settings remained unchanged. While this approach was able to maintain the tidal volume in Patient 1, Patient 2 received insufficient tidal volume with a maximal deficit of 173 ml (35%) in the extreme A–D case (figure 4 and table 3). Such discrepancies in tidal volume also occur when pairing patients with varying ARDS (B–C, B–D and C–D; table 3).

## 3.3. Testing a simple design for equilibrating tidal volumes when ventilating two patients with mismatched lung compliance

To achieve balanced tidal volumes between mismatched patients, we propose the inclusion of a variable flow restrictor (modelled as a variable resistor) in the inspiration arm of the splitter for both

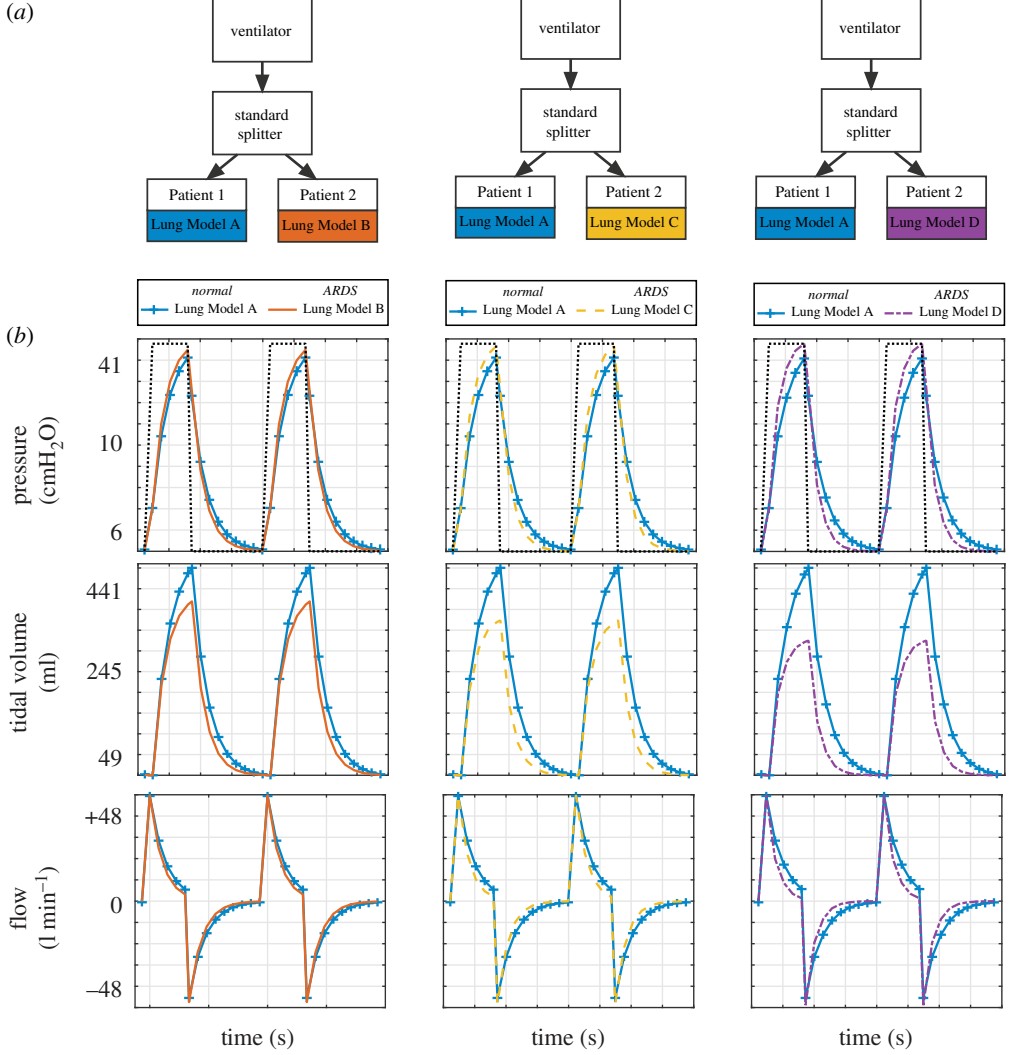

**Figure 4.** Comparison of pressure–time, volume–time and flow–time curves for pairs of patients with dissimilar lung models. (*a*) In this experiment, two patients are connected via the standard splitter to the ventilator. For each simulation, patients in each pair were parametrized with mismatched lung models, i.e. Lung Model A + Lung Model B (left column); Lung Model A + Lung Model C (middle column); Lung Model A + Lung Model D (right column). (*b*) Ventilation was performed in pressure control mode with PIP 15cmH$_2$O, PEEP 5 cmH$_2$O, RR 15 breaths/min, I : E ratio 1 : 2 (dotted line, top panel). Pressure–time, tidal volume–time and flow–time graphs are shown in each panel for Patient 1 (blue) and Patient 2 (orange, yellow or purple). Note that tidal volume for Patient 2 decreases as lung compliance decreases while tidal volume for Patient 1 remains constant.

**Table 3.** Achieved tidal volumes during pressure control ventilation for paired patients with mismatched lung compliance using the standard splitter and PIP was adjusted to achieve a tidal volume of approximately 490 ml in Patient 1.

| Patient 1–Patient 2 | PIP (cmH$_2$O) | Patient 1 tidal volume (ml) | Patient 2 tidal volume (ml) |
|---|---|---|---|
| A–A | 15 | 490.2 | 490.2 |
| A–B | 15 | 490.3 | 410.9 |
| A–C | 15 | 490.3 | 366.2 |
| A–D | 15 | 490.3 | 318.4 |
| B–C | 17 | 493.2 | 449.5 |
| B–D | 17 | 493.2 | 382.1 |
| C–D | 18.5 | 494.5 | 429.9 |

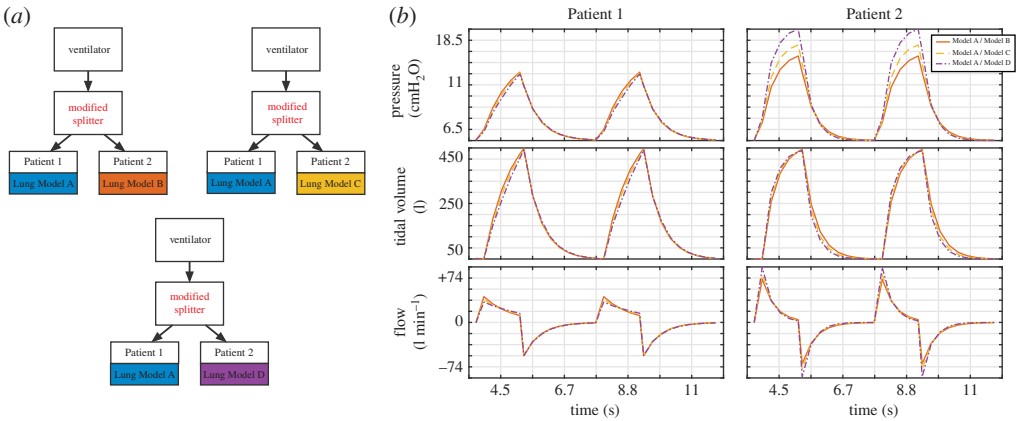

**Figure 5.** Using the modified splitter to normalize tidal volume in two patients with mismatched lung compliance. (*a*) In this experiment, two patients with mismatched respiratory compliances are connected via the modified splitter to the ventilator. The target tidal volume in all cases is that achieved when ventilating normal lungs in the model (490 ml). The following mismatched pairs were studied: Lung Model A + Lung Model B; Lung Model A + Lung Model C; Lung Model A + Lung Model D. (*b*) Pressure–time, tidal volume–time and flow–time graphs. Ventilation was performed in pressure control mode with, initially, PIP 15 cmH$_2$O, PEEP 5 cmH$_2$O, RR 15 breaths/min, I : E ratio 1 : 2 (not shown). Subsequently, PIP was progressively increased, until the target tidal volume was reached in Patient 2 with the lower compliance. At this stage, tidal volume was too large in Patient 1 (not shown) and the variable resistance in the Patient 1 inspiratory limb was therefore increased to normalize tidal volume also for Patient 1.

**Table 4.** Achieved tidal volumes during pressure control ventilation for paired patients with mismatched lung compliance using the modified splitter and with patient-specific PIP and variable resistance settings.

| Patient 1–Patient 2 | PIP (cmH$_2$O) | Patient 1 variable resistor (cmH$_2$O/L/s) | achieved tidal volume | |
| --- | --- | --- | --- | --- |
| | | | Patient 1 | Patient 2 |
| A–A | 15 | 0 | 490.2 | 490.2 |
| A–B | 17 | 7 | 496.4 | 492.9 |
| A–C | 18.33 | 11.67 | 492.9 | 488.1 |
| A–D | 20.33 | 18 | 492.5 | 488.0 |
| B–C | 18.5 | 6.67 | 492.8 | 494.2 |
| B–D | 20.3 | 11.94 | 494.0 | 494.2 |
| C–D | 20.3 | 8.67 | 494.3 | 488.1 |

patients and one-way valves (modelled as diodes) in the exhalation arm to stop pressure equilibration. We demonstrate the function of this on the cases of a patient with healthy lungs (Lung Model A) paired with a patient with increasing severity of ARDS (Lung Models B, C and D). We first show in the case of an A–A pairing that inclusion of the resistors and diodes/valves, with the resistance set to 0 recovers the standard set-up results of §3.1. We then show that by adjusting the PIP from the ventilator and altering the resistance of the resistor for the healthy patient (Patient 1 in this set-up), we can achieve tidal volume within 10 ml of 490 ml for each patient (figure 5 and table 4). This result can also be achieved when pairing patients with varying ARDS (B–C, B–D and C–D; table 4). This shows that by manipulating the inhalation pathway resistance and PIP, it is possible to equilibrate the tidal volume delivered to both Patient 1 and Patient 2 even in the setting of patients with differing lung compliances.

## 3.4. Testing a simple design for achieving different tidal volumes when ventilating two patients with matched lung disease but different tidal volume requirements

In patients with similar lung compliance, but different tidal volume requirements, it can still be necessary to achieve different tidal volumes. To show how this can be achieved when two patients are supported by

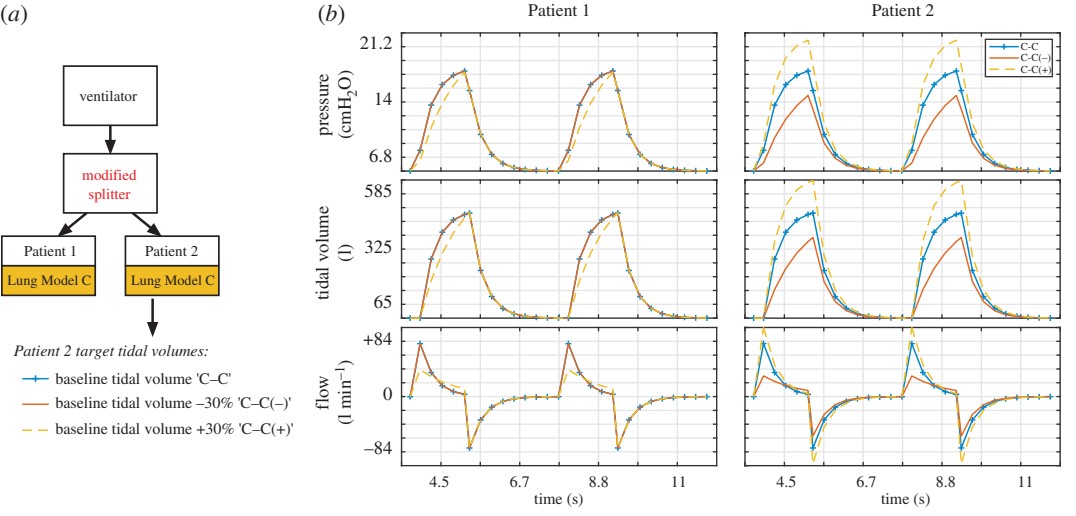

**Figure 6.** Using the modified splitter to adjust tidal volume in Patient 2 independent of tidal volume in Patient 1. (*a*) In this experiment, two patients with identical respiratory compliance (Lung Model C) are connected via the modified splitter to the ventilator. Target tidal volumes were either baseline (blue), −30% (red, C–C(−) model) or +30% (dotted yellow, C–C(+) model). (*b*) Pressure–time, tidal volume–time and flow–time graphs. Ventilation was performed in pressure control mode with PIP 17 cmH$_2$O, PEEP 5cmH$_2$O, RR 15 breaths/min, I : E ratio 1 : 2. Using these settings, a tidal volume of 493 ml was achieved for both Patient 1 and Patient 2 (blue lines). The variable resistance in the inspiratory limb supplying Patient 2 was then increased, reducing Patient 2 tidal volume while Patient 1 tidal volume remained unchanged (red lines, C–C(−) model). To increase Patient 2 tidal volume from baseline, PIP was increased to 20.67 cmH$_2$O but the variable resistance in the inspiratory limb supplying Patient 1 was also increased to return Patient 1 tidal volume to baseline (dotted yellow lines).

**Table 5.** Values of Patient 1 and Patient 2 inhalation pathway variable resistance values, ventilator PIP and resulting tidal volume for Patient 1 and Patient 2 when adjusting variable resistors and PIP to achieve equivalent tidal volume (C–C), lower tidal volume in Patient 2 (C–C(−)) and higher tidal volume in Patient 2 (C–C(+)). Tidal volume in Patient 1 is maintained throughout.

| Patient 1– Patient 2 | P1 variable resistor (cmH$_2$O/L/s) | P2 variable resistor (cmH$_2$O/L/s) | PIP (cmH$_2$O) | tidal volume (ml) | |
|---|---|---|---|---|---|
| | | | | Patient 1 | Patient 2 |
| C–C | 0 | 0 | 18.5 | 494.5 | 494.5 |
| C–C(−) | 0 | 16.67 | 18.5 | 494.2 | 379.3 |
| C–C(+) | 16.67 | 0 | 22.67 | 496.5 | 646.9 |

one ventilator using our proposed modified splitter (figure 2), we simulated two moderately severe ARDS patients (Lung Model C). We then adjusted the inhalation pathway resistors for each patient and the PIP to achieve (i) equivalent tidal volume between Patient 1 and Patient 2 (C–C); (ii) a 30% decreased tidal volume in Patient 2 (C–C(−)); or (iii) a 30% increase in tidal volume in Patient 2 (C–C(+)). We show that, by elevating the resistance in the inhalation pathway in Patient 2, we can reduce tidal volume for Patient 2, while maintaining tidal volume for Patient 1. Similarly, we show that, by increasing the resistance for Patient 1 and elevating PIP, we can maintain the tidal volume for Patient 1 and increase the tidal volume for Patient 2 (figure 6 and table 5). This shows that it is possible to independently adjust the tidal volume in Patient 2 from the tidal volume in Patient 1.

## 4. Discussion

This study does not encourage, endorse or support the use of a single ventilator to support two patients. The aim of this study was to quantify the potential risks and provide a quantitative test of a potential

solution. In this study, we have therefore (i) predicted the effects of mismatched lung compliance on achieved tidal volumes when supporting two patients from a single ventilator, and (ii) demonstrated that independent control of tidal volume can theoretically be achieved by the inclusion of variable resistance and one-way valves into the inhalation and exhalation paths of the circuit.

Our results show that when using an unmodified splitter, tidal volumes for two mismatched patients with differing lung physiology are different (figure 4 and table 3) and that it is not possible to independently control tidal volumes in these patients. Consistent with these observations, very recent technical feasibility studies of adjusting ventilator settings to achieve an optimal compromised set-up for two mismatched patients 'were not able to identify reliable settings, adjustments or any simple measures to overcome the hazards of the technique' [20]. Together these observations confirm the recent consensus statement [15] that existing equipment cannot be used to adequately ventilate two mismatched patients without an addition to the standard ventilator control system.

Here, we have proposed a closed-loop system that requires the addition of a variable resistance (modelled by a variable resistor) on the inhalation pathway. In our simulations, a decrease in compliance by 40% required the addition of a 17.35 cmH$_2$O/L/s resistance to achieve balanced tidal volumes (table 4). In this case, the total resistance to Patient 1 and Patient 2 are 25.4 cmH$_2$O/L/s and 8.05 cmH$_2$O/L/s, respectively, or approximately three times larger resistance for the healthy patient to achieve balanced tidal volumes. This compares with a 10-fold increase in resistance for a healthy lung to achieve balanced tidal volumes when a healthy lung and a diseased lung (31% decreased compliance) were paired together under a volume control ventilator set-up [20]. This discrepancy may be attributed to the way resistance was added. In a system proposed by Tronstad *et al.* [20], resistance was added to both patients' inhalation pathways, whereas in our simulations resistance was only added to the healthy patient pathway.

An open-loop variant of the solutions proposed has been very recently demonstrated in benchtop and large animal testing [21]. This system relies on a valve that releases excess volume/pressure as opposed to a variable resistor on the inspiration pathway. This is a potentially less complex system to control, but will increase the contamination risk, as compared to the closed-loop system described here. An additional consideration in this approach may be the excess loss of medical gases which could become a limiting factor in some healthcare settings [22].

## 4.1. Relevance to existing clinical guidelines

The WHO guidelines for the management of severe respiratory disease in patients with suspected COVID-19 infection recommend that, in intubated patients with ARDS, mechanical ventilation settings should be targeted to specific tidal volumes (4–8 ml kg$^{-1}$ predicted body weight) [13]. Therefore, manipulating tidal volume for individual patients, as we have demonstrated is possible with this technique, is key to these recommendations if a single ventilator/dual patient strategy is to be pursued.

Given the current interest globally on finding alternative ventilation strategies, the UK Medicines and Healthcare Products Regulatory Agency (MHRA) have released guidelines of the minimum requirement for rapidly manufactured ventilation system [23]. They state that, for the option of using pressure-controlled ventilation, both the PIP and the PEEP must be controlled. It is known that this is particularly important in this cohort of patients who will have differing lung physiology and therefore differing ventilatory requirements. In this work, we have shown that through adaptation to the conventional ventilator splitter using variable resistances it is possible to control the tidal volume to each patient through controlling the specific PIP delivered to each patient. We hypothesize that the addition of resistors on the expiratory limbs, and diodes on the inspiratory limbs, would also allow individualized control of PEEP delivered to each patient.

## 4.2. Required developments

The adoption of one ventilator to support two patients as a last resort would require several key developments, three of which are outlined below. Crucially, monitoring of both patients independently will be essential and strategies to implement monitoring and alarm systems at scale will be required. Second, technical translation of these proposed and preliminarily tested [21] solutions into clinically acceptable solutions that can be delivered at scale, delivered rapidly and once in place are easy to maintain (given the inherent staff shortages) is required. This will require simple, robust designs with limited/no connections, moving parts or electronics and designs that rely only on available equipment or components within an intensive care unit or parts that can be produced,

perhaps via three-dimensional printed, locally. Third, the development of standard protocols for controlling/calibrating the systems to deliver desired tidal volumes and pressures to each patient is required. To this end, it might be necessary to add check valves directly after the variable resistances in order to prevent flow from one patient to the other; it is possible that cases exist where there is flow from one patient to the other if check valves are not inserted. Though these check valves would involve adding components, it significantly simplifies the analysis and design of control protocols. Future work would need to establish if control protocols can be developed to allow selection of the appropriate variable resistances/PIP combinations to achieve target tidal volume for both patients, or whether additional patient monitoring is required to facilitate this proposed solution. Finally, the process for triaging, selecting and consenting pairs of patients to be supported by one ventilator needs to be established. This consideration raises ethical issues. In a ventilator-limited healthcare system the decision protocols for therapy selection need to be considered urgently, alongside the technical developments outlined above, such that appropriate clinical implementation strategies encompassing both engineering and clinical solutions could be proposed in a timely manner.

## 5. Limitations

There are several limitations of the current work. First, given that the I : E ratio and respiratory rate would need to be identical for both patients, it is likely to be necessary to cohort patients based on body weight, lung disease severity and required ventilatory parameters. Such cohorting is likely to reduce the factor by which split ventilation could increase ventilatory capacity and would need to be considered in the design of clinical protocols seeking to implement this strategy. Second, we have assumed in this model that small airways resistance remains constant under disease conditions. This is a simplifying assumption. While inspiratory resistance is only minimally increased in ARDS compared to normal subjects, increased expiration resistance has been observed [24]. Further model development using flow directional resistances in the patient model would be required to capture this complexity. Finally, the assumption of Poiseulle flow within the ventilator tubing is an approximation, and at higher flow rates, it is possible that flow becomes turbulent, especially given that the tubing is often ribbed. Future work should consider this, and/or the inclusion of nonlinear, e.g. quadratic, resistances.

## 6. Conclusion

Supporting two patients from a single ventilator is currently untested and the work presented here does not change the current clinical recommendations relating to this approach. If this approach is attempted, however, then we have shown that a simple standard set-up delivering the same PIP to both patients will result in the desired tidal volumes only if patients have the same predicted body weight and respiratory compliance. If patients are mismatched, then the difference between the achieved and target tidal volumes in our simulations was as great as 35%. The inclusion of a variable resistance and check valves into the inhalation and expiration arms of the splitter, respectively, is shown to allow the tidal volume to be set for each patient independently of their respiratory compliance.

Data accessibility. The Simulink models used in this research have been uploaded to a Git repository at the following address: https://github.com/splitvent/splitvent/releases/tag/v0.2, under release version v. 0.2. The code is available in MathWorks FileExchange platform (https://uk.mathworks.com/matlabcentral/fileexchange/75074-splitvent) and in Zenodo (doi:10.5281/zenodo.3974748). The data necessary to generate figures 3–6(*b*) has been made available as electronic supplementary material to this article in a CSV file format.
Authors' contributions. J.A.S.-L. was involved in code implementation and performing simulations. E.C. and D.D. were involved in study motivation, experimental design and writing the paper. E.C.K. was involved in model development, code verification, study design and paper writing. C.H.K. and F.T. were involved in study motivation, experimental design and writing the paper. S.N., P.E.V. and S.E.W. were involved study motivation, model design, experimental design and writing the paper.
Competing interests. We declare we have no competing interests.
Funding. This work was supported by the Wellcome/EPSRC Centre for Medical Engineering [WT 203148/Z/16/Z].
Acknowledgements. The authors would like to thank Professor Alexander Hammers, Professor Daniel Ennis and Tyler Cork for helpful discussion of the concept and Professor Alexander Hammers for critical review of the manuscript.

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
