## [Reviewer comments · Royal Society Open Science]

Review History

RSOS-200585.R0 (Original submission)

Review form: Reviewer 1

Is the manuscript scientifically sound in its present form?

Yes

Are the interpretations and conclusions justified by the results?

Yes

Is the language acceptable?

Yes

Do you have any ethical concerns with this paper?

No

Have you any concerns about statistical analyses in this paper?

No

Recommendation?

Accept with minor revision (please list in comments)

Comments to the Author(s)

The authors describe a numerical model of ventilation of two patients using a single ventilator with modification to account for potential differences in lung compliance between the two patients. The model uses an electrical analogue approach to model the system. A standard splitter and modified splitter configuration are examined and compared. The modified splitter includes variable resistors and check valves to allow targeting of specific tidal volumes during mechanical ventilation of two patients with different lung compliance.

The manuscript is generally clearly written with appropriate figures provided. Some details are missing and some aspects of the work are not discussed in the current version. The following issues should be addressed:

The current approach does not allow either respiratory rate or I:E ratio to be set independently for each paper, the authors should comment on whether this presents any issues.

Poiseuille assumptions have been used to provide resistance values. What is R_e value at the flow rates and for the dimensions of tubing in question? Is it reasonable to apply Poiseuille eqn. To determine these values, i.e. laminar conditions?

The patient model assumes no change in resistance in the presence of disease, the authors should comment on the appropriateness of this assumption.

There is no description in figure 1 or 2 or the methods of the role of the sensors in the Simulink model and how these are used, e.g. is this purely to provide output data for monitoring, or is this data also used to inform operation of the model at future time steps (e.g. feedback processes)?

Do the authors expect any variations to be observed if considering B-C, B-D, C-D configurations of patients which are not simulated in the paper currently?

No details of the solution timestep used for the numerical scheme are given? How does this relate to the time step used to post-process output data, which appears to be relatively coarse from the plots (e.g. Figure 6) where outputs have markers at intervals of ~ 0.3 seconds?

Several references refer to news articles and other documents and appear to be rather sparsely referenced, specific dates are not provided and it would be helpful to provide URLs to avoid ambiguity in locating these documents.

Review form: Reviewer 2 (Aaron B. Morton)

Is the manuscript scientifically sound in its present form?

Yes

Are the interpretations and conclusions justified by the results?

Yes

Is the language acceptable?

Yes

Do you have any ethical concerns with this paper?

No

Have you any concerns about statistical analyses in this paper?

No

Recommendation?

Accept as is

Comments to the Author(s)

Summary

The study entitled "A Simulated Single Ventilator / Dual Patient Ventilation Strategy for Acute Respiratory Distress Syndrome During the COVID-19 Pandemic" by Solis-Lemus and colleague addresses a critical shortcoming in former modeling investigations of multi patient ventilation. In particular, former modeling of multi patient ventilation assume equal lung physiology between all patients and, therefore, are likely irrelevant in a clinical setting. Indeed, mechanical ventilation (MV) use is broad, encompassing spinal cord injury, drug overdose, heart failure, and acute respiratory distress syndrome (ARDS). Patients from many of these conditions are likely to present with co-morbidities as well, making the matching of lung physiology between patients especially difficult. With this in mind, the authors a) present two electronic circuits (analogous in theory to the closed circuit of a single ventilator and two independent patients), b) created two mathematical models (standard splitter and modified splitter), c) demonstrate and quantify the pitfalls of previous multi patient MV models, d) provide a simple solution to the issues (variable flow restrictors added into the inspiration arms to control pressure for each patient and one-way valves on the expiration arms to prevent backflow), and e) made this software code freely available to the public for further testing and experimentation. Applying this model to a more clinically relevant model of multi patient ventilation, that is, pairing patients with different lung compliances on a single ventilator, the authors demonstrate that tidal volume could differ between patients as much as 38%. The authors then demonstrate through the simple addition of variable flow restrictors in the inspiratory tubes and one-way valves in the expiratory tubes, that it may be possible to ventilate two patients with mismatched lung physiology on a single ventilator.

The paper is well written, and the logic is sound. Due to the high potential need of this work and the absence of major or minor issues, we recommend this paper for publication in Royal Society Open Science.

General Comments

We want to thank the authors for performing such an important work at this time. Also, we appreciate the analogous effort of utilizing electronic circuit theory to represent respiratory physiology and the open access of these mathematical models.

Major Comments

none

Minor Comments

None

Decision letter (RSOS-200585.R0)

Dear Dr Niederer

On behalf of the Editors, I am pleased to inform you that your Manuscript RSOS-200585 entitled "A Simulated Single Ventilator / Dual Patient Ventilation Strategy for Acute Respiratory Distress

Syndrome During the COVID-19 Pandemic" has been accepted for publication in Royal Society Open Science subject to minor revision in accordance with the referee suggestions. Please find the referees' comments at the end of this email.

The reviewers and handling editors have recommended publication, but also suggest some minor revisions to your manuscript. Therefore, I invite you to respond to the comments and revise your manuscript.

- Ethics statement

- Data accessibility

If you wish to submit your supporting data or code to Dryad (<http://datadryad.org/>), or modify your current submission to dryad, please use the following link:
<http://datadryad.org/submit?journalID=RSOS&manu=RSOS-200585>

- Competing interests

- Authors' contributions

- Acknowledgements

- Funding statement

Because the schedule for publication is very tight, it is a condition of publication that you submit the revised version of your manuscript before 07-Aug-2020. Please note that the revision deadline will expire at 00.00am on this date. If you do not think you will be able to meet this date please let me know immediately.

Finally, please ensure that email addresses for all co-authors are up-to-date. At present, the following email address appears to be incorrect:

d.taylor@imperial.ac.uk

Supplementary files will be published alongside the paper on the journal website and posted on the online figshare repository (<https://rs.figshare.com/>). The heading and legend provided for each supplementary file during the submission process will be used to create the figshare page, so please ensure these are accurate and informative so that your files can be found in searches.

Files on figshare will be made available approximately one week before the accompanying article so that the supplementary material can be attributed a unique DOI.

If your manuscript is newly submitted and subsequently accepted for publication, you will be asked to pay the article processing charge, unless you request a waiver and this is approved by Royal Society Publishing. You can find out more about the charges at <https://royalsocietypublishing.org/rsos/charges>. Should you have any queries, please contact openscience@royalsociety.org.

Kind regards,
Lianne Parkhouse
Editorial Coordinator
Royal Society Open Science
openscience@royalsociety.org

on behalf of the Associate Editor and Professor R. Kerry Rowe (Subject Editor)
openscience@royalsociety.org

Associate Editor Comments to Author:

Thank you for submitting your research to Royal Society Open Science. Firstly, we apologise for the delays incurred during the initial peer-review process. We have now received the reviewer reports and we would be glad to accept your manuscript subject to the minor revisions requested by the referees. In your revised submission, please ensure to provide a point-by-point response to their comments, and also provide a tracked changes version of your paper to highlight the changes made. Lastly, we ask that you please archive your GitHub code within the Zenodo repository: <https://guides.github.com/activities/citable-code/>. Please then ensure that the Zenodo DOI is added to the data accessibility statement in addition to the GitHub URL

We look forward to receiving your revised submission.

Reviewer comments to Author:

Reviewer: 1
Comments to the Author(s)

The authors describe a numerical model of ventilation of two patients using a single ventilator with modification to account for potential differences in lung compliance between the two patients. The model uses an electrical analogue approach to model the system. A standard splitter and modified splitter configuration are examined and compared. The modified splitter includes variable resistors and check valves to allow targeting of specific tidal volumes during mechanical ventilation of two patients with different lung compliance.

The manuscript is generally clearly written with appropriate figures provided. Some details are missing and some aspects of the work are not discussed in the current version. The following issues should be addressed:

The current approach does not allow either respiratory rate or I:E ratio to be set independently for each paper, the authors should comment on whether this presents any issues.

Poiseuille assumptions have been used to provide resistance values. What is R_e value at the flow rates and for the dimensions of tubing in question? Is it reasonable to apply Poiseuille eqn. To determine these values, i.e. laminar conditions?

The patient model assumes no change in resistance in the presence of disease, the authors should comment on the appropriateness of this assumption.

There is no description in figure 1 or 2 or the methods of the role of the sensors in the Simullink model and how these are used, e.g. is this purely to provide output data for monitoring, or is this data also used to inform operation of the model at future time steps (e.g. feedback processes)?

Do the authors expect any variations to be observed if considering B-C, B-D, C-D configurations of patients which are not simulated in the paper currently?

No details of the solution timestep used for the numerical scheme are given? How does this relate to the time step used to post-process output data, which appears to be relatively coarse from the plots (e.g. Figure 6) where outputs have markers at intervals of ~ 0.3 seconds?

Several references refer to news articles and other documents and appear to be rather sparsely referenced, specific dates are not provided and it would be helpful to provide URLs to avoid ambiguity in locating these documents.

Reviewer: 2

Comments to the Author(s)

Summary

The study entitled "A Simulated Single Ventilator / Dual Patient Ventilation Strategy for Acute Respiratory Distress Syndrome During the COVID-19 Pandemic" by Solis-Lemus and colleague addresses a critical shortcoming in former modeling investigations of multi patient ventilation. In particular, former modeling of multi patient ventilation assume equal lung physiology between all patients and, therefore, are likely irrelevant in a clinical setting. Indeed, mechanical ventilation (MV) use is broad, encompassing spinal cord injury, drug overdose, heart failure, and acute respiratory distress syndrome (ARDS). Patients from many of these conditions are likely to present with co-morbidities as well, making the matching of lung physiology between patients especially difficult. With this in mind, the authors a) present two electronic circuits (analogous in theory to the closed circuit of a single ventilator and two independent patients), b) created two mathematical models (standard splitter and modified splitter), c) demonstrate and quantify the pitfalls of previous multi patient MV models, d) provide a simple solution to the issues (variable flow restrictors added into the inspiration arms to control pressure for each patient and one-way valves on the expiration arms to prevent backflow), and e) made this software code freely available to the public for further testing and experimentation. Applying this model to a more clinically relevant model of multi patient ventilation, that is, pairing patients with different lung compliances on a single ventilator, the authors demonstrate that tidal volume could differ between patients as much as 38%. The authors then demonstrate through the simple addition of variable flow restrictors in the inspiratory tubes and one-way valves in the expiratory tubes, that it may be possible to ventilate two patients with mismatched lung physiology on a single ventilator.

The paper is well written, and the logic is sound. Due to the high potential need of this work and the absence of major or minor issues, we recommend this paper for publication in Royal Society Open Science.

General Comments

We want to thank the authors for performing such an important work at this time. Also, we appreciate the analogous effort of utilizing electronic circuit theory to represent respiratory physiology and the open access of these mathematical models.

Major Comments

none

Minor Comments

None

Author's Response to Decision Letter for (RSOS-200585.R0)

See Appendix A.

Decision letter (RSOS-200585.R1)

Dear Dr Niederer,

It is a pleasure to accept your manuscript entitled "A Simulated Single Ventilator / Dual Patient Ventilation Strategy for Acute Respiratory Distress Syndrome During the COVID-19 Pandemic" in its current form for publication in Royal Society Open Science.

Please ensure that an alternative email address is supplied for Dr Taylor, as d.taylor@imperial.ac.uk is not accepting messages from the Royal Society.

COVID-19 rapid publication process: We are taking steps to expedite the publication of research relevant to the pandemic. If you wish, you can opt to have your paper published as soon as it is ready, rather than waiting for it to be published the scheduled Wednesday.

This means your paper will not be included in the weekly media round-up which the Society sends to journalists ahead of publication. However, it will appear in the COVID-19 Publishing Collection which journalists will be directed to each week (<https://royalsocietypublishing.org/topic/special-collections/novel-coronavirus-outbreak>)

If you wish to have your paper published immediately please notify production@royalsociety.org and press@royalsociety.org when you respond to this email.

You can expect to receive a proof of your article in the near future. Please contact the editorial office (openscience_proofs@royalsociety.org) and the production office (openscience@royalsociety.org) to let us know if you are likely to be away from e-mail contact -- if

you are going to be away, please nominate a co-author (if available) to manage the proofing process, and ensure they are copied into your email to the journal.

on behalf of Prof R. Kerry Rowe (Subject Editor)
openscience@royalsociety.org

Appendix A

We would like to thank the reviewers and editors for their positive comments and constructive comments. We have addressed each of these in the document below. We have included the initial editor and reviewer comments in black. Responses are in blue and any changes to the text are in italics and indented.

Associate Editor Comments to Author:

Thank you for submitting your research to Royal Society Open Science. Firstly, we apologise for the delays incurred during the initial peer-review process. We have now received the reviewer reports and we would be glad to accept your manuscript subject to the minor revisions requested by the referees. In your revised submission, please ensure to provide a point-by-point response to their comments, and also provide a tracked changes version of your paper to highlight the changes made. Lastly, we ask that you please archive your GitHub code within the Zenodo repository: <https://guides.github.com/activities/citable-code/>. Please then ensure that the Zenodo DOI is added to the data accessibility statement in addition to the GitHub URL

We have now updated our Github repository and created a version 0.2 of the code that we have uploaded onto Zenodo and we have updated the reference to the Zenodo repository in the text.

Reviewer: 1

The authors describe a numerical model of ventilation of two patients using a single ventilator with modification to account for potential differences in lung compliance between the two patients. The model uses an electrical analogue approach to model the system. A standard splitter and modified splitter configuration are examined and compared. The modified splitter includes variable resistors and check valves to allow targeting of specific tidal volumes during mechanical ventilation of two patients with different lung compliance.

The manuscript is generally clearly written with appropriate figures provided. Some details are missing and some aspects of the work are not discussed in the current version. The following issues should be addressed:

1. The current approach does not allow either respiratory rate or I:E ratio to be set independently for each paper, the authors should comment on whether this presents any issues.

We thank the Reviewer for highlighting this important point. Indeed, if two patients are to be ventilated from a single ventilator, then by necessity the I:E ratio and the respiratory rate must be identical for both patients. In the context of Acute Respiratory Distress Syndrome, I:E ratios may often be reduced from normal and variations in respiratory rate may need to be considered. However, when designing this study we felt that one reasonable approach could involve cohorting patients into groups based on body weight, disease severity and required I:E ratio/respiratory rate. Owing to this requirement for cohorting, the factor by which split ventilation would increase ventilatory capacity is likely to be less than 2. Ongoing research in our group is seeking to retrospectively analyse ventilatory requirements amongst Covid patients in order to estimate this factor; but the results will not be available until later in the year. In the meantime we have added the following paragraph to the manuscript explaining this limitation:

Given that the I:E ratio and respiratory rate would need to be identical for both patients, it will likely be necessary to cohort patients based on body weight, lung disease severity and required ventilatory parameters. Such cohorting is likely to reduce the factor by which split

ventilation could increase ventilatory capacity and would need to be considered in the design of clinical protocols seeking to implement this strategy.

2. Poiseuille assumptions have been used to provide resistance values. What is Re value at the flow rates and for the dimensions of tubing in question? Is it reasonable to apply Poiseuille eqn. To determine these values, i.e. laminar conditions?

Poiseuille flow is indeed an approximation, and at higher flow rates it is possible that flow is actually turbulent, especially given the fact tubing is often ribbed. Future work should consider this, and/or the inclusion of non-linear e.g. quadratic resistances. However, even with these modifications, tubing resistances are still likely to be orders of magnitude smaller than the resistances of the other components in the circuit, and hence should not change the over nature of the results. We have added a section to the new Limitations paragraph outlining the above considerations:

Finally, the assumption of Poiseuille flow within the ventilator tubing is an approximation, and at higher flow rates it is possible that flow becomes turbulent, especially given that the tubing is often ribbed. Future work should consider this, and/or the inclusion of non-linear e.g. quadratic resistances.

3. The patient model assumes no change in resistance in the presence of disease, the authors should comment on the appropriateness of this assumption.

We again thank the Reviewer for raising this important point. Increase in respiratory resistance may occur owing to patient factors or ventilatory circuit factors. In ARDS, inspiratory resistance is only slightly higher than in normal subjects and a significant increase in respiratory resistance may suggest ventilatory circuit factors (e.g. a blocked endotracheal tube). Whilst we have not simulated such a scenario in the present manuscript; the model as provided does allow for analysis of this situation which would be simulated by increasing the values of R_{ETT1} and R_{ETT2} . The model as provided also allows for analysis of the effects of increased small airways resistance which would be simulated by increasing the value R_{LX} in the patient submodules. As the model is currently implemented it does not, however, allow for the scenario where inspiratory and expiratory resistance are different, which may occur in ARDS. However, future development of the proposed model could include the addition of two resistances and two diodes within the patient module which would allow this more complicated scenario to be studied. Rather than conduct extensive simulations of every such scenario we preferred in the present manuscript to describe the model and publish the source code to make the model available to the research community. We have added a section to the new Limitations paragraph outlining the above considerations:

... we have assumed in this model that small airways resistance remains constant under disease conditions. This is a simplifying assumption. Whilst inspiratory resistance is only minimally increased in ARDS compared to normal subjects, increased expiration resistance has been observed. Further model development using flow directional resistances in the patient model would be required to capture this complexity.

4. There is no description in figure 1 or 2 or the methods of the role of the sensors in the Simulink model and how these are used, e.g. is this purely to provide output data for monitoring, or is this data also used to inform operation of the model at future time steps (e.g. feedback processes)?

In the manuscript, “sensors” are Simscape blocks used to denote which signals should be logged for analysis, plotting, etc. These do not represent actual sensors or model any physical aspects of real sensors. We have omitted these blocks in the figures to make for a less cluttered figure. In a clinical setting we would, of course, expect appropriate pressure and flow sensors to be in place for monitoring of the patient, which should then be used by the clinician to decide whether any settings should be adjusted as by the clinician. We have added the following text to the caption of Figure 1 to clarify what is meant with “sensor”:

The text “sensor” on the patient blocks indicate that the respective signals are connected to appropriate Simscape sensor blocks, namely current and voltage sensors, to allow for these signals to be logged during simulation.

5. Do the authors expect any variations to be observed if considering B-C, B-D, C-D configurations of patients which are not simulated in the paper currently?

We ran the model across a wide range of combinations, including those suggested above. In all cases trends were similar to those presented for the combinations analysed in the initial manuscript. We have now included these results in Table 4 for completeness and noted that they give the same results as the healthy – ARDS pairing simulations.

6. No details of the solution timestep used for the numerical scheme are given? How does this relate to the time step used to post-process output data, which appears to be relatively coarse from the plots (e.g. Figure 6) where outputs have markers at intervals of ~ 0.3 seconds?

After extensive analysis, we were satisfied with the default Simulink and Simscape solver settings. We therefore decided to use the default settings for the presentation of the results and release of the code on GitHub. The key default solver selection parameters are: “Variable-step” and “Automatic solver selection”. The following relevant settings are set to “auto” by default: max step size, min step size, initial step size, and absolute tolerance. The relative tolerance is set by default to 10^{-3} . The algorithm within Simulink ended up choosing the solver “ode45” with max step size 0.3. The time steps used to post-process data are the same as those generated by the solver. Since we have provided all files via GitHub, the readers will be free to change the max step size or other settings, should they wish to plot the data at a finer scale. We have added the following text to the Implementation section:

The default Simulink and Simscape settings were used to generate the data, namely automatic selection of solver and step sizes with a relative error tolerance of 10^{-3} . Simulink/Simscape was able to reduce the model to an ODE and chose to solve the ODE with the Dormand-Prince explicit, variable-step size Runge-Kutta (4,5) formula (the Matlab ode45 solver), with maximum step size of 0.3. The plots were generated using the resulting variable-step size data, but joined up with lines.

7. Several references refer to news articles and other documents and appear to be rather sparsely referenced, specific dates are not provided and it would be helpful to provide URLs to avoid ambiguity in locating these documents.

We thank the Review for highlighting these ambiguities. URLs have been added where relevant.

Reviewer: 2

Comments to the Author(s)

Summary

The study entitled “A Simulated Single Ventilator / Dual Patient Ventilation Strategy for Acute Respiratory Distress Syndrome During the COVID-19 Pandemic” by Solis-Lemus and colleague addresses a critical shortcoming in former modeling investigations of multi patient ventilation. In particular, former modeling of multi patient ventilation assume equal lung physiology between all patients and, therefore, are likely irrelevant in a clinical setting. Indeed, mechanical ventilation (MV) use is broad, encompassing spinal cord injury, drug overdose, heart failure, and acute respiratory distress syndrome (ARDS). Patients from many of these conditions are likely to present with co-morbidities as well, making the matching of lung physiology between patients especially difficult. With this in mind, the authors a) present two electronic circuits (analogous in theory to the closed circuit of a single ventilator and two independent patients), b) created two mathematical models (standard splitter and modified splitter), c) demonstrate and quantify the pitfalls of previous multi patient MV models, d) provide a simple solution to the issues (variable flow restrictors added into the inspiration arms to control pressure for each patient and one-way valves on the expiration arms to prevent backflow), and e) made this software code freely available to the public for further testing and experimentation. Applying this model to a more clinically relevant model of multi patient ventilation, that is, pairing patients with different lung compliances on a single ventilator, the authors demonstrate that tidal volume could differ between patients as much as 38%. The authors then demonstrate through the simple addition of variable flow restrictors in the inspiratory tubes and one-way valves in the expiratory tubes, that it may be possible to ventilate two patients with mismatched lung physiology on a single ventilator.

The paper is well written, and the logic is sound. Due to the high potential need of this work and the absence of major or minor issues, we recommend this paper for publication in Royal Society Open Science.

General Comments

We want to thank the authors for performing such an important work at this time. Also, we appreciate the analogous effort of utilizing electronic circuit theory to represent respiratory physiology and the open access of these mathematical models.

Major Comments

none

Minor Comments

None

We thank the Reviewer for his or her supportive comments about our manuscript.